# Nutrition, Bioenergetics, and Metabolic Syndrome

**DOI:** 10.3390/nu12092785

**Published:** 2020-09-11

**Authors:** Francesc Josep García-García, Anna Monistrol-Mula, Francesc Cardellach, Glòria Garrabou

**Affiliations:** 1Muscle Research and Mitochondrial Function Laboratory, CELLEX-IDIBAPS, Internal Medicine Department, Faculty of Medicine, University of Barcelona, Hospital Clinic of Barcelona, 08036 Barcelona, Spain; fjgarcia@ub.edu (F.J.G.-G.); annamoni95@gmail.com (A.M.-M.); fcardell@clinic.cat (F.C.); 2CIBERER—Centre for Biomedical Research Network in Rare Diseases, 28029 Madrid, Spain

**Keywords:** mitochondria, metabolic syndrome, mitochondrial dysfunction, balanced diet, nutrients, lifestyle

## Abstract

According to the World Health Organization (WHO), the global nutrition report shows that whilst part of the world’s population starves, the other part suffers from obesity and associated complications. A balanced diet counterparts these extreme conditions with the proper proportion, composition, quantity, and presence of macronutrients, micronutrients, and bioactive compounds. However, little is known on the way these components exert any influence on our health. These nutrients aiming to feed our bodies, our tissues, and our cells, first need to reach mitochondria, where they are decomposed into CO_2_ and H_2_O to obtain energy. Mitochondria are the powerhouse of the cell and mainly responsible for nutrients metabolism, but they are also the main source of oxidative stress and cell death by apoptosis. Unappropriated nutrients may support mitochondrial to become the Trojan horse in the cell. This review aims to provide an approach to the role that some nutrients exert on mitochondria as a major contributor to high prevalent Western conditions including metabolic syndrome (MetS), a constellation of pathologic conditions which promotes type II diabetes and cardiovascular risk. Clinical and experimental data extracted from in vitro animal and cell models further demonstrated in patients, support the idea that a balanced diet, in a healthy lifestyle context, promotes proper bioenergetic and mitochondrial function, becoming the best medicine to prevent the onset and progression of MetS. Any advance in the prevention and management of these prevalent complications help to face these challenging global health problems, by ameliorating the quality of life of patients and reducing the associated sociosanitary burden.

## 1. Introduction

In the last decades, global dietary patterns have experienced a transition towards increasingly Westernized and less healthier diets. This trend is linked to the substantial increase in the prevalence of non-communicable diseases (NCDs) such as type II diabetes (T2D) or cardiovascular disease (CVD) [1]. During the last 40 years, the global prevalence of T2D has almost doubled, from 4.7% in 1980 to 8.5% in 2014. Likewise, the number of deaths attributed to CVD (mainly coronary heart disease, stroke, and rheumatic heart disease) has increased from 14.4 million in 1990 to 17.5 million, being the number 1 cause of death globally [2].

The metabolic syndrome (MetS), a cluster of pathologic conditions that raise the risk of CVD and T2D [3], has been emerged as a health problem in modern society, associated with enormous social, personal and economic burden in both the developing and developed world [4]. Currently, MetS is calculated to affect 25% of the global population [5] and all the epidemiological studies have demonstrated its prevalence increases with age [6]. Additionally, the growing age of the population in first world societies will increase such trends in coming years.

International food organizations have traditionally focused on food security and micronutrient deficiency. However, diet-related health burdens contributing to the above-mentioned diseases are now surpassing those due to undernutrition in nearly every region of the world. Therefore, the establishment of healthy dietary patterns is a global priority to reduce the onset of these diseases [7].

There are plenty of studies establishing healthy dietary patterns [1]. However, scarce knowledge is available on the way these nutrients underscore or promote our health. First, because there is a wide panoply of nutrients and, second, because each one of them can trigger a multitude of effects, organ, tissue, cell, and even molecule specific. However, a universal fact is that all of them need to be processed into small subunits and metabolized by mitochondria to obtain energy. Thus, mitochondrial function is key to understand dietary effects in our organism and, at the same time, understanding how dietary patterns and nutrients influence mitochondrial function is essential to establish causal relationships between food and health.

The diet’s impact on mitochondrial function has been widely studied due to its role as a powerhouse of the cell and its involvement in nutrients’ metabolism. In fact, mitochondria are critical in the maintenance of metabolic flexibility, efficient switches in metabolism depending on the environmental demand (feeding/fasting cycles) [8]. Moreover, also the intake amount (p.e. high-fat and/or sugars diet) conditionate mitochondrial function [9]. Additionally, micronutrients (the ~40 essential vitamins, minerals, fatty acids, and amino acids) and bioactive compounds, must be considered in a balanced diet to get an optimal cell–mitochondrial–metabolic axis function [10].

These dietary considerations are closely related to the manifestation of the most common NCDs [11], including MetS [12], where the mitochondrial dys(function) can be considered a common hallmark.

Consequently, the aim of this work is to review the role of some nutrients, easily incorporable in a balanced diet, over mitochondrial (dys)function and its impact on MetS. For this purpose, we herein have reviewed data extracted from experimental and clinical studies published in PubMed and Google Scholar database, from inception to July 2020, to document the principal examples of nutrients’ effect on mitochondrial activity and their influence on MetS.

## 2. A Healthy Diet

Diet is the most significant single risk factor for disability and premature death [13]. Dietary choices contribute to the development of pathologic conditions such as hypertension, hypercholesterolemia, overweight/obesity, and inflammation, which in turn increase the risk for associated diseases, including CVD or T2D. Several studies have demonstrated the causal link between the global shift towards Westernized dietary patterns, characterized by high intakes of red and processed meat, pre-packaged foods, sweets, and refined grains, and lacking in fruits and vegetables, whole-grains, fish, and nuts, and the increased prevalence of these diet-related chronic diseases in the last decades [1,14].

Dietary changes recommended by the World Health Organization (WHO) include balancing energy intake, limiting saturated and trans fats, and shifting toward unsaturated fats consumption, increasing intake of fruits and vegetables, and limiting the consumption of sugar and salt. These dietary patterns are naturally occurring in certain world regions, as it is the case of the Mediterranean diet (MedDiet) and traditional Asian diets such as Korean, Chinese, and Japanese diets. Both the MedDiet (rich in fish, monounsaturated fats from olive oil, fruits, vegetables, whole grains, and legumes/nuts) and traditional Asian diets (composed of rice and other whole grains, fermented food, soy, fruits and vegetables, fish and seafood) [1] have moderate to strong evidence for preventing T2D and MetS, decreasing cancer and CVD incidence and mortality, decreasing overall mortality, and treating obesity [15,16,17,18]. Moreover, healthy dietary patterns have also been developed based on nutrient intake studies and their outcomes, as it is the case of DASH (Dietary Approaches to Stop Hypertension) [19] and MIND (Mediterranean-DASH Intervention for Neurodegenerative Delay) diets [20], among others.

At the molecular level, a healthy diet is characterized by the consumption of appropriate proportions of macronutrients (carbohydrates, proteins, and fats) to support energetic and physiologic functions, sufficient micronutrients (vitamins and mineral) and hydration, to meet the physiologic needs of the body [1]. Additionally, there is increasing evidence of the beneficial role of bioactive compounds, such as polyphenols or lycopene, in a healthy diet to prevent diseases. These molecules, generally natural and non-nutritive compounds, can be found in small quantities in plants and lipid-rich foods, and are essential to fight age-related and chronic diseases [21].

## 3. Metabolic Syndrome (MetS)

The MetS is not a disease per se, but a constellation of pathologic conditions that raises the risk of CVD by 1.5–2.5 fold [22] and T2D by 3–20 fold, depending on the numbers of pathologic factors [23]. To be diagnosed with MetS, at least three of the five deregulations shown in Table 1 must be present in the patient [24].

The determining factors for the development of MetS are, in order of relevance/influence: weight, genetics, aging, and lifestyle [25]. Based on that, risk factors associated with MetS can be reduced by the introduction of a balanced-diet and some regular exercise [26].

## 4. Mitochondria in Health and Disease

Mitochondria are unique structures in the cell, originally alpha-proteobacterium phagocyted by a eukaryotic progenitor [27]. Since then, proper mitochondrial activity is critical for the survival of the eukaryotic cell. In fact, mitochondrial deregulation has been described as a hallmark underlying several complex diseases, since mitochondrial dysfunction can trigger, by its own, oxidative stress, bioenergetic failure, inflammation, protein accumulation, and cell death [28], common aspects in metabolic disorders, such as MetS (Figure 1). That is why the nature of mitochondrial disorders underlies the numerous roles that mitochondria play.

### 4.1. Mitochondrial (Dys)Functions

The best-known mitochondrial function is energy production, through the generation of ATP molecules, via oxidative phosphorylation. The mitochondrial matrix houses the Krebs cycle (KC) enzymes, important producers of electron donors such as NADH and FADH_2_. These molecules bring electrons to the mitochondrial respiratory chain (MRC), formed by four complexes (I-IV), that pump protons from the mitochondrial matrix into the inner membrane space through sequential redox reactions that finally reduce O_2_ into H_2_O at complex IV and, the complex V or ATP synthase, uses the accumulated proto-gradient energy to phosphorylate ADP into ATP (chemiosmotic theory of Mitchell).

One of the major cell producers of NADH is the oxidation of fatty acids (FA) to acyl-CoA, a process called β-oxidation, which takes place in the mitochondrial matrix. The acetyl-CoA resulting from this process can enter the KC and get oxidized, coupled to the production of reducing power [29]. Furthermore, alterations in these metabolic routes have also been widely described in cell and animal models, as well as in patients suffering the most prevalent NCDs such as heart failure, T2D [30], or MetS [31,32].

Secondarily, these redox reactions generate reactive oxygen species (ROS) (Figure 2). Low ROS increase can trigger healthy physiological responses by activating complex transcriptional cascades of stress resistance, cell proliferation, and differentiation pathways [33]. However, when cells overproduce ROS, those can trigger pathologic consequences by promoting damage of cellular compounds (in sugars, lipids, proteins, or nucleic acids) [34,35], thus impairing cell proliferation, differentiation, and apoptosis, among others [36]. Additionally, mitochondria can trigger or amplify by their own cell death signals triggered by an alternative source of damage. Altogether, these effects may have a deep impact in mitochondrial deregulation and derived diseases, including MetS [37].

To fight oxidative stress, understood as the imbalance between ROS production and antioxidants in favor of ROS species, cells have several molecules involved in the antioxidant defense system. The enzymatic barrier includes superoxide dismutase (SOD), catalase (CAT), glutathione reductase (GR), and glutathione peroxidase (GPx). Whilst other barriers, non-enzymatic antioxidants molecules, are vitamins E and C, glutathione (GSH), and various carotenoids and flavonoids [4], present in foods and beverages.

The proton leak generated in the MRC is mainly used to produce ATP but can also be used for thermogenesis purposes or to reduce the mitochondrial membrane potential, aiming to protect cells from oxidative stress [38]. The uncoupling proteins (UCPs) are a family of 5 proteins responsible for uncoupling and widely distributed in specific tissues. In fact, several studies in rodents and patients demonstrated abnormal expression or activity of UCPs related to obesity [39] and T2D [40].

Interestingly, mitochondrial functions reach far beyond bioenergetics. Mitochondria are the key organelles in iron metabolism, involved in heme and Fe-S clusters synthesis [41], essential structures in oxygen transport, MRC, and DNA repair machinery.

Moreover, mitochondria regulate Ca^2+^ homeostasis, a key factor regulating aerobic metabolism and one of the apoptosis triggering factors [42,43]. Alterations in these pathways are linked to NCDs [44,45].

For all the above mentioned, mitochondria are key organelles for cell survival and adaptation to changing environmental conditions. Thus, for instance, high energy requirements activate mitochondrial biogenesis, induce the generation of new MRC units, and trigger an integrated physiological response of metabolic and functional remodeling. Several molecules are involved in the detection of specific signals of mitochondrial activity, such as NAD^+^:NADH, AMP:ATP ratios, or acetyl-CoA levels. That is why alterations in this fragile equilibrium conduct to a dysregulated mitochondrial activity, leading to the altered pathophysiological response observed in, among others, MetS.

Similarly, mitochondrial activities depend on the abundance of these organelles, their morphology, and their functional properties, regulated, among others, by mitochondrial dynamics [46,47,48]. Mitochondria are permanently fusing and fissioning with each other, allowing the renewal of genetic, structural, and functional components. It is especially the mitochondrial fusion process, which has been reported to allow the exchange of mitochondrial content, including mitochondrial DNA (mtDNA), which may buffer partial defects and transient stresses [49,50]. The loss of fusion and fission abilities results in altered mitochondrial populations, which leads to the most common NCDs [51].

To summarize, mitochondrial dysfunction can be due to several levels of deficiency and lead to different pathological features. All their levels of function are closely related, affecting one each other, rarely having isolated effects. In the case of MetS, reduced biogenesis conducing to variations in mitochondrial number, altered membrane potential, leading to defective energy production and defects in the activity of oxidative proteins, has been reported, all together linked with the accumulation of ROS in cells and tissues [52].

### 4.2. Some Interactions with Other Cell Structures

Mitochondria are not isolated organelles inside the cell, so interactions with other structures must be considered in health and disease. For instance, interactions with lysosomes [53] allow the degradation and turnover of defective or unnecessary mitochondrial by mitophagy, important to maintain mitochondrial quality control (MQC). The failure of the MQC system has been described in NCDs patients [54,55,56], demonstrating the importance of mitochondrial interactions with other eukaryotic cell structures.

Another important interaction of mitochondria to support cell functionality is with the endoplasmic reticulum (ER) [57], also known as mitochondria-associated membranes, MAMs, which allows another level of organelle and cell regulation. Alterations in MAMs have been demonstrated in NCDs patients [58,59,60] due, in part, to increased ER stress but also to mitochondrial dysfunction.

### 4.3. Some Mitoregulators

#### 4.3.1. Epigenetics and Sirtuins

Epigenetic modifications in DNA, which include acetylation, methylation, and ubiquitination, among others [61], determine the transcriptional machinery accessibility to the genome [62]. However, these regulatory modifications are not limited to DNA and located in the nucleus, they can also target RNA, proteins, and mitochondria, and be associated with disease [63,64].

One well-known epigenome effector, which work as cellular sensors of metabolic status, are the NAD^+^ dependent histone deacetylases sirtuins [65]. Sirtuins are a family of seven proteins involved in different biological functions [66] such as DNA repair [67], apoptosis control [68,69], ROS and oxidative stress regulation [70], together with their modulatory activity over enzymes from catabolism to anabolism [71]. Some of them have mitochondrial location or activity [66,70].

The most studied sirtuins are Sirt1 and Sirt3. Genetic or pharmacological Sirt1 inhibition can induce insulin resistance (IR) [72], and genetic Sirt1 variations have been associated with human energy expenditure and obesity [73,74,75]. Moreover, the fasting state increases NAD^+^ levels and activate metabolism through Sirt3 [76]. In the absence of Sirt3, cell and mitochondrial proteins become hyperacetylated, contributing to the progression of metabolic diseases such as MetS [77,78].

#### 4.3.2. miRNAs, mitomiRs, mitoRNAs, and xenomiRs

Non-coding ribonucleic acids (ncRNAs) are a wide heterogenous group of regulatory molecules described in the last decades. Inside this group, there are micro-ribonucleic acids (miRNAS), short molecules that prevent mRNA translation or induce transcripts degradation, thus preventing the generation of the protein and associated function. These molecules are mostly located in the cytoplasm, but there is a subset of about 150 different miRNAs, named mitomiRs, that have been also found in the mitochondrial fraction [79]. Furthermore, the transcriptome of mitochondrial genome has also revealed that it encodes multiple non-coding RNAs, which are named mitoRNAs [80]. Furthermore, in the past few years, these molecules have been associated with several NCDs [81,82,83], including MetS [84]. Interestingly, dietary changes in MetS patients associated with clinical improvements have been linked to changes in miRNAs profile [85], as later depicted.

Nutrients can interplay at different levels with the human body. Their components can be the anabolic precursors for the synthesis of complex molecules needed for cell survival. They can also interplay at the chemical level with cell constituents; this is the case of the antioxidant effect. The last findings support the idea that they can also regulate the expression of our genes. This is the case with the xenomiRs, or food-derived miRNAs, exogenous miRs that emulate the function of our endogenous miRs, exerting an additional mechanism of regulation for the expression of our genes. Despite there being only few data available and plenty of polemics, some studies revealed an important concentration of rice [86] or milk [87] xenomiRs in the plasma, after a dietary intervention. On the contrary, other publications were unable to detect them [88,89]. Additionally, in case these xenomiRs were absorbed, there is the crucial need to understand if they exert any function in our cells, as endogenous miRs do. What is certain is that this field of study will be of growing interest in the next coming years. 

### 4.4. Contribution of Mitochondrial (Dys)Function to MetS

The association between MetS and mitochondrial (dys)function has been suspected since long ago and its contribution to the deterioration of the metabolic risk factors involved in the disease has been studied.

Westernized diets, that usually include high caloric intake (high fat and/or sugars) might promote an increase in the MRC activity that could lead to the overproduction of mitochondrial ROS [90]. In turn, the oxidative stress leads to reduced ATP production and decrease of the overall metabolism in the cell, which is tightly associated with MetS [91,92,93]. Moreover, oxidative stress might contribute to the appearance of insulin resistance (IR), defined as a decreased cellular response to physiological insulin levels [94]. However, it remains unclear whether mitochondrial (dys)function is the main cause of IR or vice-versa [95,96].

Furthermore, oxidative stress directly interferes in lipid oxidation, which results in an increased cellular lipid accumulation (mainly diacylglycerols and ceramides), a situation that inhibits insulin signaling [94,97,98]. This situation affects the translocation of the glucose transporter type 4 (Glut-44) in insulin-sensitive tissues such muscle or fat [99], which leads to circulating glucose accumulation and consequently, to hyperglycemia [100].

Moreover, lipids and free fatty acids (FFA) also accumulate in the circulatory system, leading to hypertriglyceridemia [101], which in turn promotes the increase of visceral adiposity and a modification of fat distribution, and consequently the increase of waist circumference. This situation is caused by the inability of adipose tissue to store the energy excess, so the FFA overload is accumulated in the muscle, heart, liver, or as subcutaneous fat [102]. Under these conditions, fat redistribution is accompanied by macrophages infiltration, which secrete proinflammatory molecules such as interleukin-6 (IL-6), tumor necrosis factor (TNF), or C reactive protein (CRP), generating a low-grade chronic inflammation state in MetS patients [103,104]. Furthermore, abdominal obesity correlates with systemic levels of oxidative stress biomarkers [105,106] and defective mitochondrial biogenesis, manifested by impaired mitochondrial dysfunction, which includes alterations in oxidative metabolism, low mitochondrial gene expression, and reduced ATP generation [107,108]. Similarly, mitochondrial alterations, increased oxidative stress, and low-grade chronic inflammation contribute to the etiology of hypertension [109,110]. Finally, oxidative stress in also one of the main triggers of the formation and progression of the atheroma plaque responsible of cardiovascular events included in the MetS.

## 5. Nutrients (Well-Being Through Feeding)

One of the best options to avoid, delay, or minimize risk factors contributing to MetS is changing lifestyle routines, including a balanced diet.

### 5.1. Monounsaturated Fatty Acids (MUFAs)

MUFAs are fats with one unsaturation in their carbon chain. Dietary MUFAs can have different origin, thus in Western diets, the supply is through foods of animal origin, whilst the main MUFAs source in south European countries is extra virgin olive oil containing oleic acid [111]. Other sources of MUFAs are the rest of vegetables oils and nuts such as macadamia nuts, hazelnuts, or pecans [112].

#### 5.1.1. Some Extra Virgin Olive Oil Everywhere. Oleic Acid

Some cellular model studies have reported that oleic acid can have a positive role in mitochondrial function. In fact, one study from in vitro modeling of β-pancreatic cells showed that oleic acid enhances antioxidative defense [113]. Moreover, alternative studies demonstrated that oleic acid supplementation reduces ROS generation and protects mitochondria from oxidative stress or apoptosis [114,115]. Furthermore, oleic acid has been also reported as an anti-inflammatory modulator through AMP-activated protein kinase (AMPK) activation and its activity among different targets like nuclear factor kB (NF-kB) inhibition, and consequently, decreasing cytokines secretion such as TNF [116,117,118].

In turn, oleic acid also enhances fatty acid β-oxidation through AMPK response and interaction with Sirt1 and the peroxisome proliferator-activated receptor gamma coactivator 1-alpha PGC-1α [119,120], the master gene regulator of mitochondrial remodeling and biogenesis. Finally, oleic acid has been demonstrated to increase the carnitine palmitoyl transferase 1 (CPT-1) level promoting the fatty acid transport into mitochondria for β-oxidation [121,122]. Of note, most of these mechanisms have been described using in vitro models responding to higher concentrations of oleic acid compared to those obtained by food intake. Interestingly, several clinical benefits have been described with lower doses (see Section 5.1.2). Further and deeper investigation is needed to validate the mechanisms and dose-response effects of oleic acid, often observed in proof-of-concept studies.

#### 5.1.2. Extra Virgin Olive Oil Against MetS

Systematic review and meta-analysis of cohort studies have demonstrated the positive effects of oleic acid decreasing mortality risk (11%) and cardiovascular risk (12%) [123,124]. Focusing attention on MetS, dietary MUFAs showed similar results [112]. In fact, clinical trials have demonstrated a reduction of MetS risk factors such as blood pressure or total low-density lipoprotein (LDL) cholesterol [125]. Moreover, also an improvement on insulin sensitivity and inflammatory response have been demonstrated using oleic acid in cell models [126] and randomized clinical trials [127]. Finally, recent systematic review on human intervention studies with enriched oleic acid diets demonstrated that this intervention can reduce central obesity and abdominal fat [128].

### 5.2. Polyunsaturated Fatty Acids (PUFAs)

PUFAs are fats with more than two unsaturations in their carbon chain that can be divided into two main groups. Omega-3 fatty acids (α-linolenic acid, eicosapentaenoic acid and docosahexaenoic acid) mainly found in blue fish, eggs, and walnuts. In turn, the omega-6 group (linoleic acid, arachidonic acid, and docosapentaenoic acid) are mostly founds in vegetable oils, such as sunflower, soybean, corn, and canola oils.

#### 5.2.1. More Blue Fish and Less Vegetal Oils. Omega-3 vs. Omega 6

Omega-3 PUFAs have been reported to act as mitochondrial protectors, as they reduce ROS production and inflammation through the activation of AMPK. In turn, AMPK activates PGC-1α which, together with the decrease in fission genes such as dynamin related protein 1 (DRP1), stimulates mitochondrial function and fusion processes [129]. PGC-1α triggers different signaling pathways that promote mitochondrial biogenesis, β-oxidation, glucose utilization, antioxidants detoxification, and the activation of uncoupling proteins, which leads to a decrease in lipid accumulation and a reduction of ROS [130,131,132,133]. It has also been described to attenuate ER-stress and the subsequent disruption of Ca^2+^ homoeostasis. All these actions decrease the expression of pro-inflammatory cytokines and attenuate the inflammasome, which improves insulin sensitivity [134]. Some animal studies and clinical trials demonstrate that these actions would be triggered by changes in miRNAs profiles, among others [135,136]. In addition, the reduction of the omega-6/omega-3 balance and the omega-3 incorporation into mitochondrial membranes increase its viscosity. This situation improves mitochondrial function and avoids cell dysfunction and cell death [137]. However, the disbalanced omega-3/omega-6 ratio in detriment of omega-6 may also lead to adverse consequences, since excessive omega-3 PUFAs may eventually trigger pro-oxidant and pro-inflammatory environments [138]. Thus, the conservation of the omega-3/omega-6 ratio balance is key to avoid detrimental effects (Figure 3). 

#### 5.2.2. Omega-3 PUFAs Against MetS

It is still controversial whether omega-6 PUFAs exert pro-inflammatory or anti-inflammatory effects by themselves [139]. However, it is known that an unbalanced omega-6/omega-3 ratio in favor of omega-6 PUFAs is highly prothrombotic and proinflammatory. The high amounts of omega-6 PUFAs present in Western diets, together with the very low amounts of omega-3 PUFAs, leads to an unhealthy omega-6/omega-3 ratio that can reach a 20:1 proportion (instead the expected 1:1) [140], contributing to atherosclerosis, obesity, and T2D [141] (Figure 4).

Moreover, animal studies have shown that omega-3 improves insulin sensitivity and reduces visceral fat storage [142,143]. Different randomized clinical trials have proven that omega-3 supplementation produces favorable hypolipidemic effects, a reduction in pro-inflammatory cytokine levels and improvement in glycaemia [144]. Animal and patient-based studies have shown that high omega-3 intake and low omega-6/omega-3 ratio are associated with lower MetS risk [145,146].

### 5.3. Vitamins

Vitamins are essential micronutrients for the proper functioning of metabolism. However, vitamins cannot be synthesized by the organism, or not in enough amount, so they must be obtained through diet. Vitamins can be classified into hydrosoluble (vitamins B and C) and liposoluble (vitamins A, D, E, and K). Their biological properties will be discussed below.

#### 5.3.1. A Varied Diet. Vitamins B

Vitamins B comprise a set of molecules found in a wide variety of food groups such as dairy products (B2, B3, B5), eggs (B1, B3, B7, B12), fish and meat (B1, B3, B5,B6, B12), or plant-based foods (B1, B5, B6, B9). Vitamins B are essential in the KC, so deficiencies in some of these vitamins will lead to impaired mitochondrial metabolism, increased levels of mitochondrial ROS production, and decreased overall energy production [147]. 

Vitamin B1 (thiamin) is a cofactor of several enzymes such as the cytosolic pyruvate dehydrogenase or the mitochondrial ketoglutarate dehydrogenase; in vitro models demonstrate that its deficiency is linked to increased ROS formation [148]. Vitamin B2 (riboflavin) is essential as a prosthetic group for several enzymes catalyzing redox reactions; different studies demonstrated that its deficiency is associated with a loss of mitochondrial complex IV and induced oxidative stress [149]. Vitamin B3 (niacin) is an important cofactor involved in mitochondrial respiration reactions, glycolysis, or lipid oxidation; B3 have also demonstrated to have an important antioxidant activity in different investigations [150]. Vitamin B5 (pantothenic acid) is an important prosthetic group (CoA) involved in KC and lipid metabolism; cellular models also demonstrated an important role in the enzymatic antioxidant defense system promoting CAT, GR, or GPx [151]. Vitamin B6 (pyridoxal) have several functions in different metabolic pathways such one-carbon reactions, gluconeogenesis, or lipid metabolism, among others; even have an important antioxidant role through its activity in glutathione pathways, demonstrated in cellular models [152]. Vitamin B7 (biotin) is an important cofactor for some carboxylase enzymes involved in lipid metabolism; decreased B7 levels have been linked to increased ROS formation and impaired mitochondrial respiration [153]. Vitamin B9 (folic acid) exerts different biological functions like nucleotide synthesis or mitochondrial tRNA modification, among others; folate contributes to cellular redox states preventing oxidative stress indirectly [154]. Finally, vitamin B12 (cobalamin) is essential for nucleotide synthesis or succinyl-CoA generation; B12 deficiency has been related to indirect increased ROS levels due to glutathione dysregulation in patients [155].

#### 5.3.2. Orange Juice. Vitamin C

Vitamin C, or ascorbic acid, is principally found in pepper (and other vegetables) and fruits (mainly in kiwi, guava, and citrus). This molecule is a potent antioxidant that activates Sirt1, triggering a signaling cascade that decrease ROS and apoptosis and also acts as a ROS quencher. Moreover, it is also involved in carnitine biosynthesis, the key factor in β-oxidation. Thus, vitamin C deficiency will lead to impaired ATP production, β-oxidation deficiency, and ROS formation [156,157], and therefore, mitochondrial dysfunction.

#### 5.3.3. Carrot Cream. Vitamin A 

Vitamin A or retinol is mainly found in carrots, sweet potatoes, or spinaches, as well as pork and beef liver, which concentrate the highest amounts. This vitamin affects mitochondrial function as it enhances the levels of the mitochondrial transcription factor A (mtTFA), a key transcription factor for mitochondrial function, and plays a key role in glycolytic energy generation, as an essential cofactor for protein kinase C delta (PKCẟ), which signals the pyruvate dehydrogenase complex for an enhanced flux of pyruvate into the KC [158]. Thus, deficiency of vitamin A has been associated with decreased respiration and ATP synthesis [158,159,160]. 

#### 5.3.4. Slight Daily Sunbathe. Vitamin D

Vitamin D is naturally present in blue fish and egg yolks, among other foods. Furthermore, vitamin D can be synthesized by the human body when ultraviolet rays from sunlight strike the skin. However, after its obtention, either from sun exposure or food, it needs two hydroxylations to be biologically active.

Many biological functions of vitamin D are mediated by the control of the vitamin D receptor (VDR) on nuclear transcription. The VDR main function is to regulate Ca^2+^ homeostasis through the upregulation of calcium transporters. Moreover, VDR protects from an excessive respiratory activity and limits ROS production by controlling mitochondrial and nuclear transcription of the proteins involved in MRC and ATP synthesis [161,162]. 

#### 5.3.5. Avocado Is the Answer. Vitamin E

Vitamin E is mainly found in avocado, leafy vegetables such as spinach and broccoli, sunflower seeds, nuts, and olive oil. Vitamin E is the most important antioxidant in cell membranes. It protects mitochondrial structure and function by maintaining mitochondrial membrane integrity, allowing the recovery of respiratory function and reducing lipid and protein oxidation [163]. This protective function would be, in part, explained by the inhibition of lipid peroxidation and the increase in CAT, SOD, GR, and GPx activity demonstrated in T2D rats [164].

#### 5.3.6. Vitamins Against MetS

The role of vitamins in mitochondrial metabolism and their antioxidant properties suggest that they might play a role in the prevention of MetS. Indeed, a case control-study showed that plasma levels of vitamins A, C, E, and D were significantly lower in MetS patients compared to healthy subjects [165,166].

Different animal studies have suggested that vitamin A supplementation ameliorates obesity through UPC1-mediated thermogenesis and delays the appearance of T2D by increasing mtTFA [158,167]. Vitamin B7, biotin deficiency has been linked to impaired glucose tolerance, hyperglycemia, and decreased glucose oxidation in animal studies [168] and a cross-sectional study showed that high levels of vitamin B12 are protective to MetS [169]. Regarding vitamin C, different patient-based studies have suggested that intake or supplementation decreases the risk of MetS and improves the quality of life of MetS patients [169,170], especially when combined with physical exercise [171,172]. Moreover, a cross-sectional study with more than 2000 patients demonstrated that deficiency of vitamin C was associated with an increased likelihood of MetS [169]. Several epidemiologic studies have established a strong association between vitamin D deficiency and hypertension, obesity, and dyslipidemia [173,174]. In a randomized clinical trial with MetS patients, vitamin E was seen to exert beneficial effects on cytokines and the lipid profile [175]. Moreover, different studies have found that vitamin E could ameliorate the pathologic conditions associated with MetS such as hyperglycemia or obesity, although more studies in patients are needed to prove the consistency of these findings [175,176].

Nevertheless, when it comes to vitamins, overconsumption may be just as bad as scarcity. For instance, an excess of vitamin A intake (with a tolerable upper intake level for adults of 3000 μg/day) [177] promotes oxidative stress and mitochondrial death [178] and it is associated with an increased likelihood of MetS [160]. A high intake of vitamin B3 (niacin), can result in niacin-induced IR [179], and when Vitamin C and E are consumed in excess, they promote prooxidants activity instead of antioxidant function [147].

### 5.4. Trace Elements

Trace elements are minerals present in small amounts in living tissues. Some of them are nutritionally essential and so need to be incorporated through diet, such as selenium or zinc, and others are considered nonessential (or there is not consistent evidence to regard them as essentials) [180]. 

#### 5.4.1. A Handful of Nuts. Selenium and Zinc

Selenium and zinc are mainly present in nuts, mushrooms, fish, selfish, and meat, with antioxidant and mitochondriogenic properties. Selenium promotes mitochondrial biogenesis through the stimulation of PGC-1a, while the nuclear redox factor1 (NRF-1) generation exerts its antioxidant role activating both GPx and GR enzymes. Some of these positive effects would be modulated by selenium impact on miRNAs profile preventing oxidative stress and inflammation [181]. On the other hand, zinc exerts its antioxidant power by promoting SOD and CAT activation, inhibiting important pro-oxidant enzymes such as NADPH oxidase and competing with redox-active transition metals such as iron and copper for certain binding sites (cell membranes, proteins), thereby prohibiting them from catalyzing the formation of ROS and the initiation of lipid peroxidation [182].

#### 5.4.2. Trace Elements Against MetS

Some studies have shown that adequate intake of these trace elements is associated with an attenuation of MetS risk factors. In vivo and patient-based studies have found that recommended doses of selenium (1–2 μg/kg/day) act as an insulin-mimetic, attenuating diabetes [183,184,185] and have cardioprotective effects (increase plasma antioxidant capacity and decrease lipid peroxidation and LDL-cholesterol) [186,187,188]. Additionally, a cross-sectional study including more than 2000 patients showed that dietary selenium intake was negatively associated with MetS [170]. Regarding zinc, it is involved in different protective mechanisms against obesity, IR, hypertension, and dyslipidemia, which suggests potential roles of zinc in both the prevention and treatment of MetS. However, the current scientific evidence in humans regarding the associations between zinc status and occurrence of MetS is still inconsistent [189].

Despite these healthful and wellness properties of minerals, as it happens with vitamins, excess of these elements is just as bad as a deficiency. Fortunately, severe impact on human health of vitamins and minerals is only reached at very high concentrations. For instance, doses of selenium 100 times higher than expected had adverse effects than at recommended doses by increasing the risk of T2D and promoting CVD [190,191,192].

### 5.5. Polyphenols

Polyphenols are bioactive compounds with antioxidant properties mostly found in plant-based foods [193]. According to their molecular structure, polyphenols can be categorized into two main groups: flavonoids (such as catechins) and non-flavonoids, including phenolic acids (such as ellagic acid), stilbenes (such as resveratrol), lignans (such as pinoresinol), and others (in which we find oleuropein and hydroxytyrosol). Many of them show mitochondrial effects and have been reported to modulate MetS [194].

#### 5.5.1. A Cup of Green Tea. Catechins

Inside the flavonoids family, we find the catechins molecules, present mainly in green tea (with a total minimum content of catechins of 8%), but also in a wide variety of fruits (Table 2). It has been demonstrated that catechins can directly work as ROS scavengers and metal ions chelators and activate some antioxidant enzymes such as CAT and SOD. Moreover, they can also inhibit some pro-oxidant enzymes like NADPH-oxidase and suppress stress-related signaling pathways (as those mediated by TNF or the activator protein 1). Some in vitro studies also demonstrated an anti-inflammatory effect of catechins through changing miRNAs profile [195]. All together, these effects contribute to decrease oxidative stress and improve mitochondrial function [196].

#### 5.5.2. Some Black Grapes. Resveratrol

Resveratrol is another antioxidant, mainly found in black grapes. Resveratrol is a mitochondrial protective agent. It activates Sirt1 which triggers different signaling pathways that promote antioxidant and anti-inflammatory effects, and parallelly activate PGC-1α, the master regulator of mitochondrial biogenesis. Sirt1 activation enhances β-oxidation and glucose utilization, promotes mitochondrial biogenesis, antioxidant detoxification, and the expression of uncoupling proteins [233,234]. Moreover, some of the anti-inflammatory effects of resveratrol are triggered by change in miRNAs profiles, as demonstrated in cell models [235] and clinical human trials [236]. Of note, resveratrol is one of those nutrients were herein reported beneficial in vitro effects have been observed with concentrations usually beyond those reached by food intake. Notwithstanding, and regardless the mechanism, clinically beneficial effects have also been reported [236]. Further studies are required to deepen in dose-response effects of resveratrol, their interaction with other nutrients, and host metabolism.

#### 5.5.3. A Spoon of Olive Oil. Oleuropein, Hydroxytyrosol, and Pinoresinol

Olive oil have a great content of polyphenols such as oleuropein (OL), hydroxytyrosol (HT), and pinoresinol. Olive oil polyphenols (OOPs) are mitochondrial-protective agents for its antioxidant properties. It has lately emerged that OOPs, particularly HT, are able to induce Sirt1 expression. Moreover, Sirt1 interaction with nuclear redox factor 2 (NRF2) signaling, responsible for the transcriptional activation of antistress target genes, exerts a protective effect against oxidative stress. Furthermore, OL and HT can also scavenge ROS and OOPS have been demonstrated to activate antioxidant enzymes such as GPx and CAT. Finally, HT has also been reported to stabilize some mitochondrial complex-subunits and enhance the expression of PGC-1α and CPT-1 [204], with previously reported mitochondrial activity.

#### 5.5.4. Polyphenols Against MetS

Different studies have proven the beneficial effects of polyphenols in the prevention of MetS. Animal studies have shown that green tea catechins lowers blood pressure by reducing ROS [199] and have lipid-lowering activities [202]. Moreover, catechins and OOPs have been demonstrated to enhance glucose tolerance and decrease events related to IR in both animal and human-based studies [203,204]. In addition, cardioprotective roles have been attributed to resveratrol and OOPs thanks to their ability to counteract inflammation and reduce LDL-oxidation [204,207,208].

In humans, a meta-analysis of 20 randomized clinical trials, with a total of 1536 participants who received green tea regularly, showed a slight decrease in systolic blood pressure and a moderate reduction of LDL cholesterol [200]. Another study with 48 MetS patients demonstrated that dietary achievable doses of blueberries significantly decreased blood pressure [201]. This blood pressure-reducing effect has also been attributed to OOPs by different intervention studies [209]. Regarding resveratrol, a meta-analysis including 16 studies, 10 human-based and 6 in vivo studies, showed that resveratrol intake meaningfully reduces the body weight, waist circumference, triglycerides, and glucose level [205]. Moreover, a meta-analysis including 28 randomized clinical trials also documented an improvement in obesity measures with resveratrol supplementation [206].

### 5.6. A Tomato Salad. Lycopene with Oleic Acid

Lycopene is a carotenoid found mainly in tomato, but also in other fruits such as watermelon, papaya, or red grapefruits. The lycopene content increases during different stages of fruit ripening, so the higher content of lycopene is found in ripe tomatoes [237].

However, lycopene undergoes photo-oxidation and degradation with light, which produces a decrease in bioavailability. The incorporation of lycopene in an oil phase overcomes photo-oxidation. An interesting option is the association of lycopene with olive oil, which prevents lycopene degradation and enriches its healthy properties with oleic acid, a monounsaturated fatty acid with antioxidant properties [238,239]. 

Lycopene is also a powerful antioxidant and anti-inflammatory molecule. It acts as both ROS and nitrogen species scavenger [240], decreases DNA damage [241,242,243], and modulates the production of SOD and GPx [244,245]. Moreover, it promotes the expression of constructs containing antioxidant response elements (ARE). Regarding its anti-inflammatory effects, lycopene reduces apoptosis as well as the expression of inflammatory cytokines [246]. Lycopene also exerts lipid-lowering properties through the induction of Sirt1 activity [247].

#### Lycopene Against MetS

Based on the potent antioxidant and lipid-lowering properties of lycopene, different studies have assessed it as a beneficial nutrient for the prevention and treatment of MetS. Animal and human-based studies have shown that lycopene reduces blood pressure [210,211], atherosclerotic burden [248] and improves blood antioxidant capacity [246,249], has an anti-obesity role [212,213], improves insulin sensitivity, reduces hyperglycemia [216,217], and improves the lipid profile [214,215]. Moreover, a retrospective study with 2500 patients with MetS showed that higher serum levels of lycopene are associated with a reduced risk of death [250].

### 5.7. Garlic Seasoning Is Always a Good Idea. Organosulfur Compounds (OSCs)

The OSCs comprise a group of molecules found in garlic that are accumulated as γ-glutamyl peptides. Depending on the processing status of the garlic, different OSCs can be present: alliin and sulfur amino acids from whole cloves, allicin from crushed cloves, ajoene and vinyldithiines from stir-fried garlic, and (poly)sulfides from steam-distilled garlic oil. However, the main active metabolite responsible for most of the biological activities of garlic is allicin [251].

The OSCs are potent antioxidant molecules, they decrease the activity of pro-oxidant molecules, such as NADPH oxidase, and increase the activity of antioxidant proteins, such as NRF2 [252,253]. Moreover, they also activate Sirt3, which prevents cardiac oxidative stress and mitochondrial dysfunction [224]. OSCs have also anti-inflammatory effects, as they suppress the activation of proinflammatory molecules such as NF-kB or TNF [254,255].

#### OSCs against MetS

In vitro and in vivo studies have highlighted the anti-adipogenic effect of OSCs through the promotion of mitochondrial uncoupling (increasing the expression of UPC-1) and the inhibition of adipocyte differentiation (through AMPK activation) [218,219,220,221], as well as their cardioprotective effects (decrease plasmatic insulin, total cholesterol, and oxidative stress levels) [224,225]. The cardioprotective effects of garlic have also been demonstrated in patient-based studies [226].

Moreover, different intervention studies have shown that garlic supplementation attenuates MetS abnormalities: reduces hypertension [222,227,256], improves lipid profile [222,223], decreases waist circumference and fasting blood glucose [222].

## 6. The Impact of Gut Microbiome on MetS. between Nutrients and Mitochondria

The human intestines are home to a vast number of bacteria, archaea, microbial eukaryotes, and viruses, collectively known as the gut microbiome [257]. Nutrients can modify microbiome composition (this is the case of yogurts and pickles) and, at the same time, microbiome can condition the absorption of some nutrients and, thus, our health. That is why gut microbiome can be considered another intrinsic factor contributing to MetS, and many other conditions.

Diet is accepted as the first driver of gut microbiome composition and not only the cause of preventable diseases. In 2007, it was demonstrated that a high-fat-diet affects the intestinal barrier leading the pass of microbiota products to systemic circulation. These molecules, such as lipopolysaccharide (LPS), showed deleterious effects on glycaemic function and activated the inflammatory response [258]. Recently, LPS has been also related to plasmatic decrease in high-density lipoprotein (HDL) cholesterol and increase in triglycerides [259,260].

In fact, the gut microbiome responds to diet composition [261], including changes depending on fat composition [262,263], fiber types [264,265], and food additives [266]. Moreov73er, some dietary interventions demonstrated changes in the microbiome population that negatively correlated with obesity and hypertension [267,268,269]. Some others do not. This is the case of dietary interventions with polyphenols in MetS patients, that despite decreasing the expression of MetS biomarkers, yielded to null differences in gut microbiota when compared to healthy subjects [270].

Furthermore, also gut microbiome metabolic end products like the short-chain fatty acids (SCFAs) that may play a role in MetS. Some SCFAs like butyrate appear diminished in inflammatory diseases [271,272], whilst another SCFAs study with propionate supplementation improved visceral adipose tissue levels and maintained insulin sensitivity [273]. Additionally, when produced in high amounts, these molecules have been also involved in obesity, among others [274]. 

Finally, gut microbiome can synthesize vitamin K, necessary for normal clotting activity, and some B vitamins, contributing to more than a quarter of the daily reference intake in the case of B3, B6, B9, and B12 [275]. All these vitamins are cofactors of some metabolic routes involved in proper mitochondrial function. 

Beyond food and food compounds, some other interventions have an impact on gut microbiome and, consequently, on host metabolism, such as prebiotics, probiotics, and also fecal microbiome transplantation (FMT). Prebiotics are defined as non-digestible polysaccharides that promote “the selective stimulation of growth and/or activity(ies) of one or a limited number of microbial genus(era)/species in the gut microbiota that confer(s) health benefit to the host” [276]. In turn, probiotics are defined as “live microorganisms which when administered in adequate amounts, confer a beneficial health effect on the host” [277]. Several studies have been conducted in animal models [278,279,280] and humans [281,282,283,284], demonstrating benefits such as decrease in serum glucose, visceral fat area, waist circumference, or cholesterol levels. However, these results have not been always reproduced [285,286]. Finally, FMT has demonstrated that it can help reverse obesity in rodents, although there is no evidence in humans [287]. However, FMT has demonstrated to increase insulin sensitivity in MetS patients [288,289]. 

All these data suggest a role of gut microbiome in MetS, conditioning the interplay and effects of nutrient intake within the organism. However, this research area is quite novel and needs further assessment to understand the complex interactions between gut microbiome and its host.

## 7. Diet as a Therapy

### 7.1. Healthy Diet Base: Which Food and in Which Context

The nutrients and bioactive compounds reviewed above are just an example of the beneficial impact of some dietary patterns to avoid the onset and development of MetS and its associated complications. They exert a multitude of healthy actions, some of them through the modulation of mitochondrial functions.

For that purpose, dietary habits should focus on what and how we eat rather than on nutrients. This means predominating plant-based foods and fish, typical from the traditional MedDiet [290] rather than supplementation containing the nutrients or bioactive compounds (except for specific situations). Accordingly, a recent meta-analysis has shown that a prudent/healthy dietary pattern is a protective factor for MetS [291]. 

Nevertheless, multiple additional factors must be considered to achieve a health-promoting lifestyle. For instance, proper hydration it is very important. According to the European Food Safety Authority (EFSA) the adequate intake of daily total water for women and men is 2 and 2.5 L, respectively [292]. In fact, suboptimal hydration has an impact on health [293], metabolism [294] and T2D [295].

Additionally, a healthy lifestyle must minimize elements that have negative effect on health, and particularly on MetS such as tobacco [296], alcohol [297], or physical inactivity [298]. 

Moreover, related to the physical inactivity, some regular exercise has a positive impact on disease and on personal well-being [299]. Physical activity enhances energy consumption and reduces the risk of prevalent diseases such as T2D, CVD, and MetS [300]. Because of the increase of energy consumption, the metabolic disturbances associated with MetS such as increased blood glucose, triglycerides, or inflammation improve [300,301].

However, we must be careful with the intensity of physical exercise and avoid acute trainings, which has been related to an increase of ROS production [302]. For that purpose, the most beneficial practice will be moderate intensity training, promoting the proper balance between the ROS generation and the increase in the antioxidant defense [303].

### 7.2. Ketogenic Diet and Mitohormesis

Under some extreme situations, as primary mitochondrial diseases, changes in diet recommended by trained professionals may help to reconfigure mitochondrial activity and consequently, adverse clinical features [11,304].

This is the case of the ketogenic diet (KD), based on the intake of high-fat vs. low-carbohydrates content. This situation mimics starvation, forcing the body to use fat as energy source. Fatty acid oxidation results in ketogenesis [305] in the liver, with the consequent exportation of ketonic bodies throughout the rest of the body.

This diet must be prescribed and monitored by professionals to assess benefits and manage side effects [306]. Surprisingly, KD has shown anticonvulsant properties, at the clinical level, and the modulation of mitochondrial biogenesis [307] and the regulation of the levels of hormones or neurotransmitters [308], at the molecular level. These effects can be explained, at least in part, for an adaptive response called mitohormesis. Mitohormesis can be defined as the biological response where the induction of a reduced amount of mitochondrial stress leads to an increment in health and viability within a cell, tissue, or organism [309]. In this case, fat consumption would promote modest ROS increase, stimulating cell proliferation, differentiation, immunity, and adaptations that would eventually (and surprisingly) lead to enhancing resistance to oxidative stress in mitochondria [310,311].

About 25–30% of epilepsy patients worldwide show resistance to any pharmacological management (refractory epilepsy). Experiences on KD have shown a reduction in seizures greater than 50% [312,313], although the explanation for this effect remains unclear [314,315].

In inherited metabolic diseases such as Leigh syndrome, glucose transporter type 1 (Glut-1) and pyruvate dehydrogenase complex (PDH) deficiency and in a subset of patients with complex I deficiency [305,308], KD has also demonstrated to be useful. Under KD, most patients with epilepsy become seizure free and improve speech, motor, and mental development, and ability [316] despite its use is only partially able to reverse the clinical course of the neurodegenerative condition [317].

### 7.3. Enhancing MRC and Energy Buffering. Mitochondrial Burst

Some other mitochondrial diseases, contrarily, require the supplementation with CoQ10, riboflavin (vitamin B2), or thiamine (vitamin B1) to enhance ETC, or with creatine monohydrate, to increase energy buffering.

In some genetic diseases affecting enzymes involved in CoQ10 synthesis [318], supplementation can improve the clinical symptoms related to this deficiency [319]. Supplementation with riboflavin, a key building block in complex I and II, has particular usefulness in acyl-CoA dehydrogenase-9 (ACAD9) deficiency, resulting in increased complex I activity [320,321] and in the multiple acyl-CoA dehydrogenase deficiency caused by mutations in electron-transport flavoprotein dehydrogenase (ETFDH) [322,323]. Thiamine has been used, alone or in combination with other molecules, in the treatment of MELAS (mitochondrial myopathy, encephalopathy, lactic acidosis and stroke-like episodes), improving the symptoms of myopathy and lactic acidosis [324] and Leigh syndrome [325].

Finally, creatine binding to phosphate in the mitochondria is the main source of high energy in anaerobic metabolism. High energy demanding tissues like muscle or brain present the highest creatine levels [326]. In mitochondrial myopathy, the content of phosphocreatine is decreased [327], which is why supplementation with creatine monohydrate effectively improves the motor ability of patients with mitochondrial myopathy [328]. 

As explained, the complexity of dietary treatments for mitochondrial diseases requires the intervention of multidisciplinary and specialized units from clinical settings where both clinicians and dieticians design the proper treatment (including dietary interventions) for each affected patient. Although some recommendations may be similar, the dietetic support for mitochondrial diseases is set apart from the management of MetS.

## 8. Conclusions

The evidence herein discussed highlights the association of mitochondrial (dys)function with MetS. However, whether mitochondrial defects can be understood as a cause or consequence of MetS remains to be clarified. Similarly, whether all nutrients reach mitochondria and at which food dose do exert their beneficial effects is still controversial. In this sense, in vitro studies usually work with higher concentrations that significantly differ from those that are reached in human interventions and must be considered as models to understand molecular mechanisms difficult to approach in vivo. All approaches are needed to understand the interplay between nutrients, mitochondria, and MetS, or disease in general, where the effects of nutrients will differ also depending on their dose, their acute or chronic consumption, and their interaction with other food components or host characteristics. Despite growing interests, the complexity of reality is still far beyond current knowledge and information provided in the present review challenge new researchers to develop novel approaches. Whether nutrients exert their protective effect against MetS exclusively through mitochondria or by deploying a panoply of molecular effects beyond it, is also under consideration, at least for some of them.

However, some mitochondrial effects have been extensively documented in cell and animal models, as well as in patients. In particular, the above reviewed nutrients and bioactive compounds have been proved to improve mitochondrial function. In fact, oleic acid, omega-3, vitamins B and C, selenium/zinc, polyphenols, lycopene, and OSCs have been documented to improve mitochondrial antioxidant defense and reduce oxidative stress; oleic acid, lycopene, and OSCs have a potent anti-inflammatory role; vitamins A, B, and C are essential for mitochondrial energy generation; vitamin D and omega-3 regulate calcium homeostasis; vitamin E protects the mitochondrial structure; and selenium/zinc and some polyphenols stimulate mitochondrial biogenesis. At the same time, the consumption of all these elements has been associated with the reduction of MetS risk factors.

All this evidence confirms that a balanced diet represents a valuable therapeutic strategy to improve the global metabolic state, among others, through the enhancement of mitochondrial function. However, is not the intake of any of these isolated compounds by their own, as dietary supplements, but the synergic interplay of them all in foods and meals which makes the difference (for instance, lycopene of tomato with olive oil). Similarly, the excess of most of them in a diet is as bad as their absence. This is the case of the antioxidant paradox, since most of antioxidants, upon athreshold, become prooxidants compounds. In most cases, the balance is the key to obtain healthy effects within a diet. This is also the case of the intake of some foods such as fatty fish, with healthy effects for their composition in omega 3, but also prone to accumulate traces of some heavy metals and other pollutants with toxic effects at the nervous but also the cardiovascular system [329,330]. In these cases, moderate consumption is the best choice to promote healthy benefits.

Additionally, diet must not be the unique lifestyle intervention to ameliorate mitochondrial activity and MetS. In this regard, moderated physical activity has been also demonstrated to prevent and/or improve the metabolic alterations related with mitochondrial dys(function), as well as MetS, together with avoiding mitochondrial toxic drugs (as tobacco or alcohol).

In conclusion, following a balanced diet, like the MedDiet or traditional Asian diets, protects against the MetS (and many other diseases). The best medicine is in our hands; cheap, tasty, and close. However, the exact formula does not exist, maybe because it is specific for each subject. Despite all the advances in understanding the role of nutrition in mitochondrial function, the perfect dietary patterns and nutrients combination to preserve/enhance mitochondrial activity remains unclear, as well as how these dietary patterns get translated in enhanced metabolic health. For that purpose, further studies must be undertaken to fully understand the impact of a balanced diet on global/mitochondrial health. Moreover, a better comprehension of some novel intrinsic factors, such as gut microbiota interactions with nutrients and host metabolism, and extrinsic factors, such as the induced epigenetic response to dietary patterns, would contribute to the understanding of the complexity of the nutritional impact on health. Such findings should be effective for the development of the appropriate dietary interventions that may lead to avoid the onset and the development of one of the major health problems in the modern society, MetS. Interestingly, personalized medicine and individual dietary patterns will probably set the path for novel therapeutics and preventive medicine in coming years.

## Figures and Tables

**Figure 1 nutrients-12-02785-f001:**
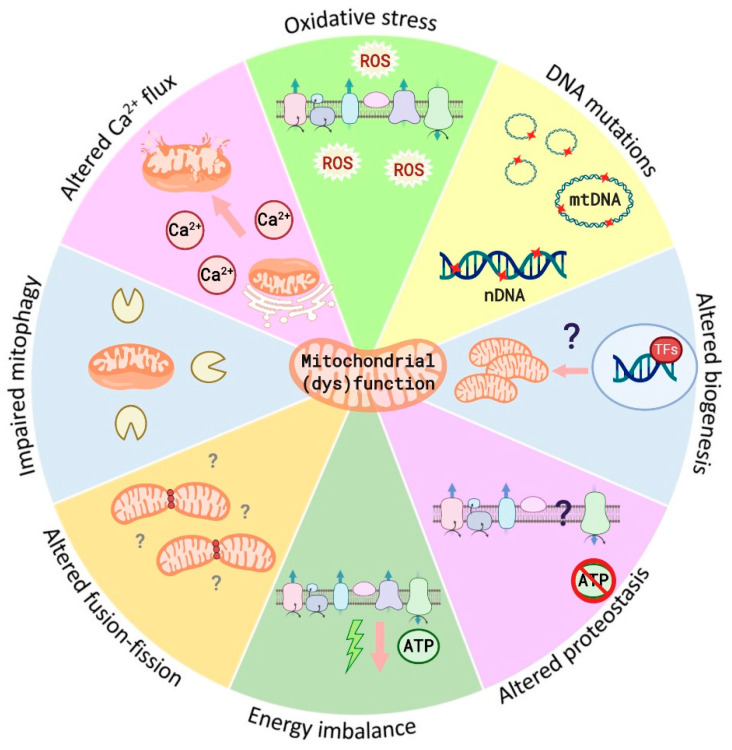
Mitochondrial (dys)function factors. Mitochondrial dysfunction is characterized by the combination of some of the processes described above and plays a central role in several diseases, such as metabolic syndrome. All these processes are closely related, usually affecting one each other and rarely having isolated effects. *Oxidative stress* leads to the accumulation of reactive oxygen species (ROS). In turn, high amount of ROS might trigger *DNA mutations.* Mitochondrial (dys)function can also be associated to *reduced mitochondrial biogenesis* through the decreased expression of mitochondrial genes. Moreover, *altered proteostasis* (biogenesis, folding, trafficking, and degradation of mitochondrial proteins) is common. Another frequent consequence is an *energy imbalance* limiting the ATP supply. Furthermore, mitochondrial (dys)function may condition mitochondrial *fusion-fission* processes compromising mitochondrial renewal and material exchange. Similarly, *impaired mitophagy* triggers the accumulation of non-functional mitochondria. Finally, mitochondrial *altered Ca^2+^ flux and buffering*, disrupt communication with other organelles such as endoplasmic reticulum and, in case of calcium release into the cytoplasm, the potential trigger of cell death.

**Figure 2 nutrients-12-02785-f002:**
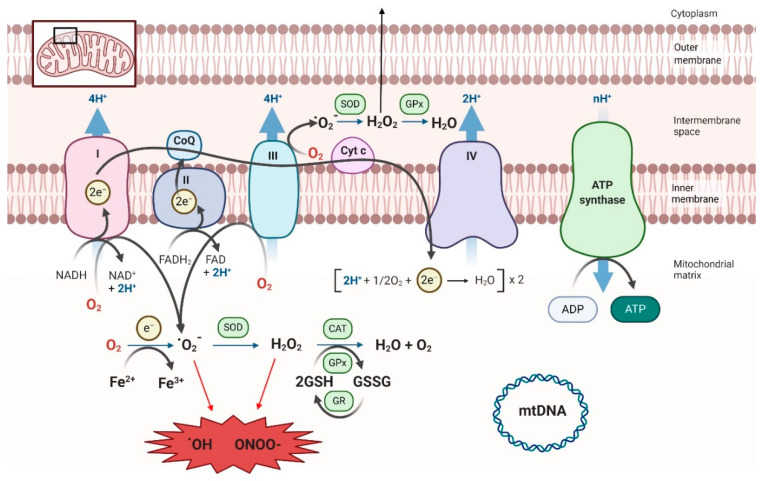
Main sources of mitochondrial reactive oxygen species (ROS). Mitochondrial respiratory chain (MRC) is composed of four enzyme complexes (I–IV). Electrons donated from NADH and FADH_2_ in the Krebs cycle are transferred at complex I and II respectively, and consecutively to complex III and IV. This electrons transfer is coupled with protons transport across the inner membrane, generating a proton-gradient, allowing ADP phosphorylation through ATP synthase (complex V). Oxygen metabolism generates superoxide anion (·0_2_^−^), which in turn is conversed into hydrogen peroxide (H_2_O_2_) by SOD (superoxide dismutase) and then converted into water thanks to the action of some of the antioxidant enzymatic defenses such as catalase (CAT) or the glutathione peroxidase (GPx) reductase (GPr) system. Excessive ROS production can oxidize proteins, lipids, or mitochondrial DNA (mtDNA), eventually leading to mitochondrial dysfunction and cell death.

**Figure 3 nutrients-12-02785-f003:**
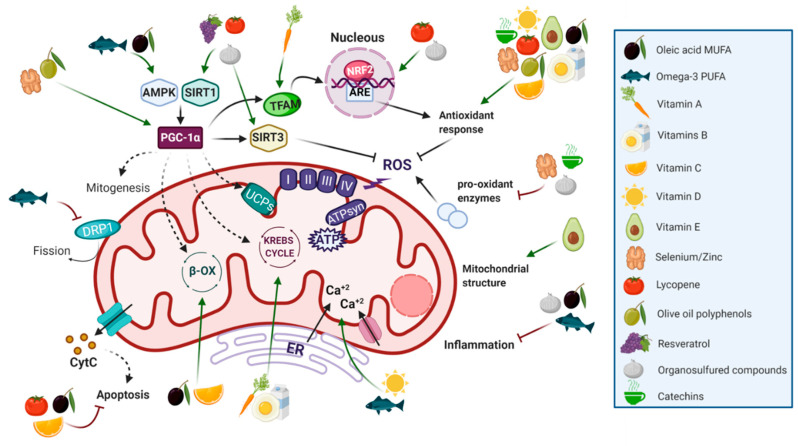
Principal nutrients with beneficial effects on mitochondrial function. Oleic acid, omega-3, selenium and zinc, resveratrol, lycopene, and the organosulfur compounds (OSCs) activate numerous pathways, among them, direct or indirectly, PGC-1α signaling, one of the main coactivators and master gene regulator of energy metabolism and mitochondrial biogenesis. The activation of the PGC-1α promotes mitogenesis, β-oxidation, glucose utilization, uncoupling, and antioxidant detoxification. Moreover, vitamin A and vitamins from the B group are essential for a correct functioning of the glycolytic energy system, and vitamin C and oleic acid for the functioning of β-oxidation, among others. One of the main effects of oleic acid, lycopene, organosulfur compounds (OSCs), vitamins B, C, D, selenium, and zinc, catechins and olive oil polyphenols (OOPs) in our cells is the decrease of oxidative stress by both promoting an antioxidant response and inhibiting pro-oxidant enzymes. Vitamin E also acts as an antioxidant and maintains a correct mitochondrial structure, while omega-3 inhibits mitochondrial fission. Omega-3 and vitamin D are involved in calcium homeostasis, among others. Vitamin C, lycopene, and oleic acid have been reported to inhibit apoptosis, while oleic acid, omega-3, and OSCs reduce inflammation.

**Figure 4 nutrients-12-02785-f004:**
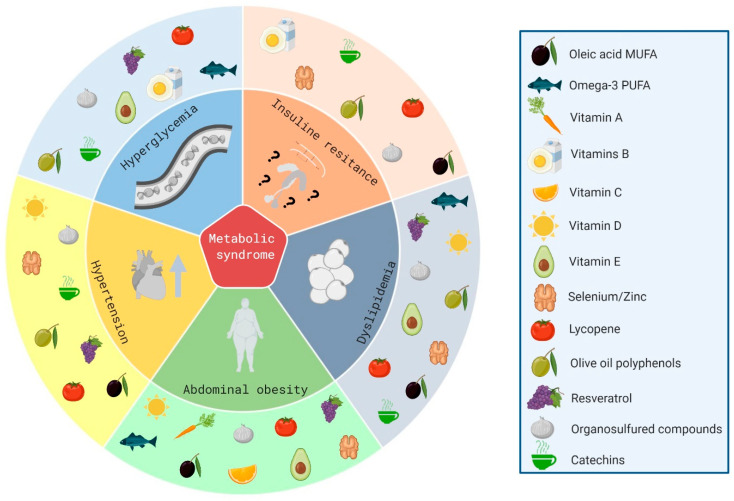
Main nutrient impact on metabolic syndrome (MetS) components. The MetS is a constellation of pathologic conditions which includes hyperglycemia, insulin resistance, dyslipidemia (hypertriglyceridemia and low high density lipoprotein (HDL)-cholesterol levels), central obesity, and hypertension. Several studies have demonstrated that nutrients such as oleic acid monounsaturated fatty acid (MUFA), omega-3 polyunsaturated fatty acids (PUFAs), vitamins A, B, C, D, and E, selenium and zinc elements, lycopene, olive oil polyphenols, resveratrol, organosulfured compounds, and catechins have a positive impact on MEtS components, improving the onset and development of the disease.

**Table 1 nutrients-12-02785-t001:** Risk factors associated with metabolic syndrome and definition of the parameters for its diagnosis.

Risk Factor	Parameters	Definition (Male/Female)
Central obesity	Waist circumference ^1^	America: ≥102 cm/≥88 cm Asia: ≥90 cm/≥80 cm Europe: ≥94 cm/≥80 cm Africa: ≥94 cm/≥80 cm Middle East: ≥94 cm/≥80 cm
Hypertriglyceridemia	Triglyceride concentrations	≥150 mg/dL
Low HDL-cholesterol levels	HDL-cholesterol levels	≤40 mg/dL/<50 mg/dL
Hypertension	Systolic/diastolic blood pressure	≥130/85 mm Hg
Hyperglycemia	Fasting plasma glucose	≥100 mg/dL

^1^ The definition of this measure varies among regions. HDL = High density lipoprotein.

**Table 2 nutrients-12-02785-t002:** Summary of the principal nutrients with beneficial mitochondrial effects influencing metabolic syndrome.

Nutrient	Molecular Group and Structure	Main Food Sources	RDA [197]	Principal Role in Mitochondria	Main Effect against Metabolic Syndrome	Main LoE	Reported References
Oleic acid	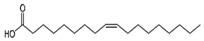 MUFA	Olive oil Vegetables oils Nuts	20 g of olive oil/day ^1^	Antioxidant Antiapoptotic Enhances β-oxidation	↓ Blood pressure	H	[125]
					Improves lipid profile	H	[125]
					↑ insulin sensitivity	H	[126,127]
					↓ inflammation	H	[126,127]
					Reduces central adiposity	H	[128]
Omega-3	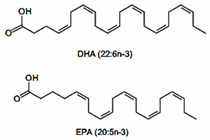 PUFA	Fish and seafood Nuts and seeds	1.1–1.6 g/day (ALA)	Uncoupling β-oxidation mitochondrial biogenesis	hypolipidemic effects	H	[140,143,144]
					↑ insulin sensitivity	H	[142,144]
					↓ inflammation	H	[140,144]
					↓ MetS risk	H	[145,146]
Vitamins B	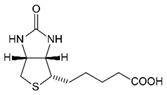 Water-soluble vitamin ^2^	Dairy products (B2, B3, B5) Eggs (B1, B3, B7, B12) Fish and meat (B1, B3, B5, B12) Plant-based foods (B1, B5, B9)	B1: 1.1–1.2 mg/day B2: 0.9–1.1 mg/day B3: 14–16 mg/day B5: 5 mg/day B6: 1.3 mg/day B7: 0.03 mg/day B9: 0.3–0.4 mg/day B12: 0.0024 mg/day	Essential in Krebs Cycle Antioxidant	Low levels are associated with hyperglycemia and insulin resistance	H	[167,179]
					Protective to MetS	H	[168]
Vitamin C	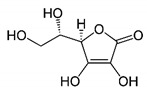 Water-soluble vitamin	Guava Citrus fruit Kiwi Strawberries Peppers	75–90 mg/day	Antioxidant Antiapoptotic Involved in β-oxidation	↑ the effects of physical activity in the prevention of MetS and the quality of life of MetS patients	H	[168,169,170,171]
Vitamin A	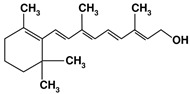 Fat-soluble vitamin	Carrots Pork/beef liver Foie Spinach Sweet potato	0.7–0.9 mg/day	Key role in mitochondrial respiration	Ameliorates obesity	A	[166]
					Delays the appearance of diabetes	A	[158]
Vitamin D	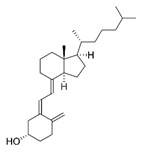 Fat-soluble vitamin ^3^	Blue fish Egg yolk Synthetized in the presence of sunlight exposure	0.015 mg/day	Controls the respiratory activity and limits ROS production	Low levels are associated with hypertension, obesity and dyslipidemia	H	[161,162]
Vitamin E	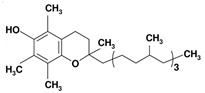 Fat-soluble vitamin	Sunflower seeds Leafy vegetables Nuts Olive oil	15 mg/day	Protects mitochondrial structure and function (antioxidant)	↓ inflammation	H	[174]
					Improves the lipid profile	H	[174]
Selenium and Zinc ^4^	NA Trace element	Nuts Fish and selfish Meat	Se: 0.055 mg/day Zn: 8–11 mg/day	Antioxidant Mitochondrial biogenesis	Insulin-mimetic	H	[183,184,185,189]
					Improves lipid profile	A	[189,198]
					Cardioprotective	A	[186,187,188]
Catechins	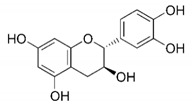 Flavonoid family	Green tea Broad beans Blueberries Black grapes Strawberries Apricots	Up to 704 mg/day ^1^	Antioxidants	↓ Blood pressure	H	[199,200,201]
					Improve lipid profile	H	[200,202]
					Insulin-like/-enhancing activities	H	[203,204]
Resveratrol	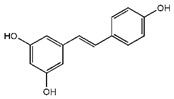 Polyphenols	Grapes Apples Blueberries Pistachios Peanuts	Up to 4 mg/day ^1^	Mitochondrial protective agent (antioxidant)	↓ body weight	H	[205,206]
					↓ waist circumference	H	[205,206]
					Improve lipid profile	H	[205]
					↓ glucose levels	H	[205]
					Cardioprotective	H	[204,207,208]
Oleuropein (OL), Hydroxytyrosol (HT), Pinoresinol	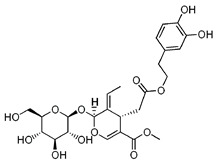 Oleuropein structure	Olive oil	20 g of olive oil/day ^1^	Mitochondrial protective agent (antioxidant)	Cardioprotective	H	[204]
					↓ Insulin resistance	H	[204]
					↓ Blood pressure	H	[209]
Lycopene	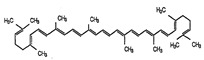 Carotenoid	Tomato Watermelon Papaya Apricots Red grapefruit Carrots Pumpkin Sweet potato	5.7–15 mg/day ^1^	Antioxidant	↓ Blood pressure	H	[210,211]
					Anti-obesogenic	A	[212,213]
					improves the lipid profile	A	[214,215]
					↑ insulin sensitivity and ↓ plasma glucose	H	[216,217]
					↓ Risk of death in MetS patients	H	[211]
Allicin	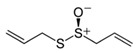 Organosulfur compounds	Garlic	1–2 g of raw garlic ^1^	Antioxidant Uncoupling	Improves obesity	H	[218,219,220,221,222]
					Improves lipid profile	H	[223]
					↓Hyperglycemia	H	[222]
					Cardioprotective	H	[224,225,226]
					↓Hypertension	H	[223,227]

^1^ Suggested recommended intake [228,229,230,231,232]. ^2^ Molecular structure from biotin (B7). ^3^ Molecular structure from cholecalciferol (D3). ^4^ The current scientific evidence in humans shows inconsistent results regarding the beneficial effects of zinc in MetS risk factors. RDA = Recommended dietary allowance; A = Animal studies; H = Human studies; NA = Not applicable.

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
