# Peer review of "Nutrition, Bioenergetics, and Metabolic Syndrome"

_nutrients, 2020, doi:10.3390/nu12092785_

Round 1
Reviewer 1 Report
This is an extremely comprehensive review and well structured. It outlines mechanisms and clearly explains links between various food compounds and metabolic syndrome.
It is however too long. It is easy to lose the plot.
Tables and figures are very useful.
Some minor comments:
please expand:
line 186: Change That's why to that is why
line 192: Change It's to it
line 197: there are too comas after MetS,,
line 249: Change What's to what is
line 570: Change That's to that is
Line 295 and 480: TNF alpha, change to TNF, now the nomenclature has changed and TNF is used. See relevant websites
Section 5.3 you list vitamins but it would be worth adding here that you will be discussing their antioxidative properties below
line 457 Zinc and line 465 Selenium- I do not think they need to be spelt with a capital letter.
Reviewer 2 Report
Aim of the review titled “ Nutrition, bioenergetics and metabolic syndrome”, is to provide an approach to the role that some nutrients exerts on mitochondria as a major contributor to high prevalent western conditions including Metabolic Syndrome “.
The goal of the review is very ambitious, and the authors put together a lot of information. However, the result of their work appears a little bit superficial. The mechanisms by which nutrients protect mitochondrial function are not always described. On the other side, also the section concerning mitochondria (4) is a little bit superficial. For example, the authors cite ROS as deleterious molecules, but ROS exert many regulatory actions in the cells. Therefore, paragraph 4 appears as a very partial list of some aspects of mitochondrial function, and their regulation without a link between the different aspects and without any deepening.
Also, section 5, concerning the effects of nutrients on mitochondrial function, appears merely descriptive. The information is reported without comment and some effects found on a specific cellular model are reported as generalized effects. This is the case of the effects of oleate that protects beta cells from ER stress (ref 114). In general, the effect of the food components on the cells is not critically evaluated. For example, is the dose of resveratrol that affects body parameters reached feeding the aliments that contain it?
In conclusion, the review notwithstanding being very interesting suffers to be too much superficial and to generalize effects found on a specific experimental model.
There are many inaccuracies for example:
Lines 219- 222 it is not clear the meaning of the sentence. ROS and oxidative stress are not cellular functions.
Line 537-537 “Moreover, a retrospective study with 2,500 MetS patients showed that higher serum levels of lycopene are associated with a reduced risk of death [226].” Patients suffering from what disease?
Line 253-254 what does it means the sentence “High caloric diets (high fat and/or sucrose) enhance the overproduction of mitochondrial ROS [91].” Do the mitochondria always overproduce ROS?
Also, English requires careful revision.
Round 2
Reviewer 2 Report
The authors responded fully satisfactorily to my observations. Therefore, I believe that the paper is adequate for publication.